# Evaluation of 16S rRNA gene sequencing for species and strain-level microbiome analysis

Jethro S. Johnson [1,7]*, Daniel J. Spakowicz [1,2,7], Bo-Young Hong[1], Lauren M. Petersen[1,3], Patrick Demkowicz [1], Lei Chen[1,4], Shana R. Leopold[1], Blake M. Hanson[1,5], Hanako O. Agresta[1], Mark Gerstein[6], Erica Sodergren[1] & George M. Weinstock[1]

The 16S rRNA gene has been a mainstay of sequence-based bacterial analysis for decades. However, high-throughput sequencing of the full gene has only recently become a realistic prospect. Here, we use in silico and sequence-based experiments to critically re-evaluate the potential of the 16S gene to provide taxonomic resolution at species and strain level. We demonstrate that targeting of 16S variable regions with short-read sequencing platforms cannot achieve the taxonomic resolution afforded by sequencing the entire (~1500 bp) gene. We further demonstrate that full-length sequencing platforms are sufficiently accurate to resolve subtle nucleotide substitutions (but not insertions/deletions) that exist between intragenomic copies of the 16S gene. In consequence, we argue that modern analysis approaches must necessarily account for intragenomic variation between 16S gene copies. In particular, we demonstrate that appropriate treatment of full-length 16S intragenomic copy variants has the potential to provide taxonomic resolution of bacterial communities at species and strain level.

[1] The Jackson Laboratory for Genomic Medicine, Farmington, CT 06032, USA. [2] Ohio State University Comprehensive Cancer Center, Columbus, OH 43210, USA. [3] Department of Pathology and Laboratory Medicine, Dartmouth-Hitchcock Medical Center, Lebanon, NH 03756, USA. [4] Shanghai Institute of Immunology, Shanghai Jaiotong University School of Medicine, Shanghai, China. [5] Center for Antimicrobial Resistance and Microbial Genomics, McGovern Medical School, Houston, TX 77030, USA. [6] Department of Computer Science, Yale University, New Haven, CT, USA. [7] These authors contributed equally: Jethro S. Johnson, Daniel J. Spakowicz. *email: Jethro.Johnson@jax.org

Since the advent of high-throughput sequencing, PCR-amplified 16S sequences have typically been clustered based on similarity to generate operational taxonomic units (OTUs) and representative OTU sequences compared with reference databases to infer likely taxonomy. Although convenient and powerful, such usage of 16S has necessitated certain assumptions, e.g., the now historic assumption that sequences of > 95% identity represent the same genus, whereas sequences of > 97% identity represent the same species[1].

16S sequences have also been exploited using low-throughput methods to distinguish strains (sometimes called subspecies) based on polymorphisms within the gene. Single-nucleotide polymorphisms (SNPs) have been used to track strains of clinical relevance or, when they are stably linked to other parts of the bacterial haplotype, to predict phenotypic characteristics[2]. Thus, accurate and complete 16S sequences are of high utility in many applications. Until recently, however, accurate, full-length 16S sequences have been beyond the scope of high-throughput sequencing platforms.

Availability of third-generation technologies means that high-throughput sequencing of the full 16S gene is becoming commonplace. Circular consensus sequencing (CCS)[3,4], combined with sophisticated denoising algorithms[5–8] to remove PCR and sequencing error, mean it is now possible to discriminate between millions of sequence reads that differ by as little as one nucleotide across the entire gene. Together, these technological and methodological advances mean that for the first time it is becoming possible to exploit the full discriminatory potential of 16S in a high-throughput manner.

Here, we demonstrate that, in the face of such changes, historical assumptions need to be revisited. Using an in-silico dataset of sequences taken from public databases we show that commonly targeted 16S sub-regions, such as V4, are unable to match the taxonomic accuracy achieved when sequencing the full 16S gene. Using long-read sequencing of mock and in-vivo communities, we demonstrate that it is possible to accurately resolve the divergent copies of the 16S gene that exist within the same genome. Finally, we demonstrate that such intragenomic 16S gene copy variants are highly prevalent in taxa isolated from the human gut microbiome, suggesting they may be used to improve discrimination between species and even strains in 16S gene-based microbiome studies.

## Results

**The full 16S gene provides better taxonomic resolution**. The ~1500 bp 16S rRNA gene comprises nine variable regions interspersed throughout the highly conserved 16S sequence (Fig. 1a). Sequencing the entire gene was originally accomplished by Sanger sequencing. This required cloning genes, generating, and assembling two to three reads per clone, and producing limited sampling depth at high cost and effort. Currently, however, the vast majority of studies sequence only part of the gene, because the widely used Illumina sequencing platform (higher throughput, lower cost, reduced effort compared with Sanger) produces short sequences ( ≤ 300 bases). Different sub-regions of the gene are therefore targeted, ranging from single variable regions, such as V4 or V6, to three variable regions, such as V1–V3 or V3–V5 (used in the Human Microbiome Project in conjunction with the 454 sequencing platform[9]).

We argue that targeting sub-regions represents a historical compromise, due to technology restrictions[10]. Today, both PacBio and Oxford Nanopore sequencing platforms are capable of routinely producing reads in excess of 1500 bp and high-throughput sequencing of the full 16S gene is becoming increasingly prevalent. We therefore suggest that the justification

for this compromise needs to be revisited and we performed a simple in-silico experiment to demonstrate the advantage of full-length 16S sequencing over the targeting of sub-regions.

We downloaded a set of non-redundant (i.e., > 1% different), full-length 16S sequences from a public database (Greengenes). Taking advantage of the fact that a substantial proportion of these sequences incorporated PCR primer-binding sites, we trimmed them to generate in-silico amplicons for different sub-regions, based on the location of PCR primers commonly used in microbiome studies (Fig. 1a and Supplementary Tables 1–2). Assuming each sequence in our downloaded database represented a unique species, we then used a common classification approach (the Ribosome Database Project (RDP) classifier[11]) to calculate the frequency with which in-silico amplicons for each sub-region could provide accurate, species-level taxonomic classification (using the original database as a reference). In a second experiment, we also clustered our in-silico amplicons to generate OTUs at different, commonly used, sequence similarity thresholds (97%, 98%, 99%).

We found that sub-regions differed substantially in the extent to which they could confidently discriminate between the full-length 16S sequences used to represent species (Fig. 1b). The V4 region performed worst, with 56% of in-silico amplicons failing to confidently match their sequence of origin at this taxonomic level. By contrast, when a full-length sequence with all variable regions was used, it was possible to classify nearly all sequences as the correct species (Supplementary Fig. 1a). Altering databases and classification confidence thresholds affected the proportion of in-silico amplicons that could be accurately matched, but did not influence prevailing trends (Supplementary Fig. 1a, b).

Second, different sub-regions showed bias in the bacterial taxa they were able to identify (Fig. 1c). For example, the V1–V2 region performed poorly at classifying sequences belonging to the phylum Proteobacteria, whereas the V3–V5 region performed poorly at classifying sequences belonging to the phylum Actinobacteria (Supplementary Fig. 2). Similar trends were seen at the genus level for taxa of potential medical relevance. Although the full V1–V9 region consistently produced the best results, the V6–V9 region was notably the best sub-region for classifying sequences belonging to the genera *Clostridium* and *Staphylococcus*, the V3–V5 region produced good results for *Klebsiella*, and the V1–V3 region produced good results for *Escherichia*/*Shigella* (Supplementary Fig. 2 and Source Data).

Finally, the choice of sub-region dramatically affected the number of OTUs formed when clustering in-silico amplicons to create OTUs. When clustering at 99% sequence identity, all sub-regions failed to recreate the number of distinct sequences present in the original database; however, the V4 region again performed worst (Fig. 1d). Notably, the relative number of OTUs produced by each sub-region was not consistent at different identity thresholds (97%, 98%, 99%, Supplementary Fig. 3), indicating that the behavior of clustering algorithms may be difficult to predict when the amount of information contained within a sequenced region is highly variable.

In conclusion, targeting sub-regions represents a historical compromise that was sufficient for identification of taxa at the genus level or above. However, our simple in-silico experiment demonstrates that it is not valid to assume that ever finer clustering of these sub-regions will result in the improved taxonomic resolution necessary to reflect species. Although some sub-regions (e.g., V1–V3) provide a reasonable approximation of 16S diversity, most do not capture sufficient sequence variation to discriminate between closely related taxa. We also note that discriminating polymorphisms may be restricted to specific variable regions; thus, certain sub-regions will be better suited for discriminating closely related members of certain taxa.

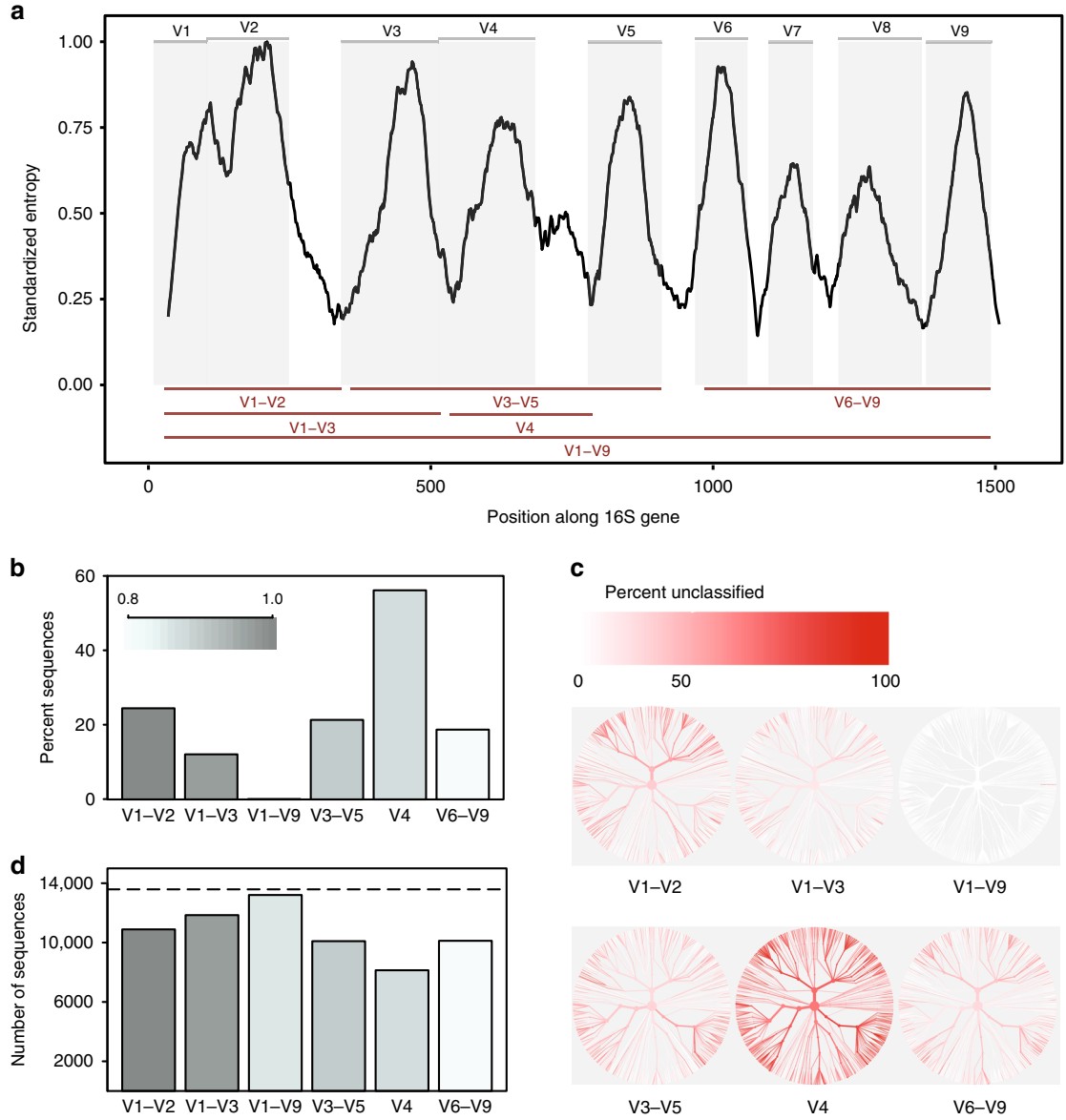

**Fig. 1** In-silico comparison of 16S rRNA variable regions. **a** Shannon entropy across the 16S gene based on the alignment of a single representative sequence for each known species present in the Greengenes database. Sequences were aligned against a single reference 16S gene for *Escherichia coli* K-12 MG1655 (NCBI Gene ID 947777). Gray panels depict variable regions defined by commonly used primer-binding sites (Supplementary Table 1). Variable regions considered in this study are shown as red lines (bottom). **b** Proportion of sequences for each variable region that could not be identified to species level when classifying each sequence against the reference database from which it was derived at a confidence threshold of 80% (RDP classifier). **c** Trees based on taxonomy of sequences present in the in-silico database. The same tree is provided for each variable region. The color of each branch reflects the proportion of sequences within each clade that could not be identified to species level. **d** The number of OTUs created when clustering sequences for each variable region at 99% sequence similarity. Dashed line indicates the number of unique sequences (>1% different) in the original database. Source data are provided as a Source Data file

**16S gene copy variants reflect strain-level variation**. Clustering of 16S sequences into OTUs has historically served two purposes. First, it has removed minor artifactual sequence variants due to PCR amplification and sequencing errors when collapsing sequences into groups. Second, it has collapsed legitimate sequence variants that exist between closely related bacterial taxa. Although the latter may not always be desirable, it stands to reason that you cannot distinguish between bacterial taxa whose 16S sequences vary at a rate that is lower than the error encountered on a particular sequencing platform.

Recently, advances in CCS have dramatically improved error rates of long-read sequencing platforms. At the same time, computational methods have made it possible to distinguish between legitimate vs. artifactual sequence variation. These technological and methodological advances mean researchers now have the potential to perform high-throughput sequencing that can accurately detect single-nucleotide variants across the entire 16S gene.

Although it is tempting to assume that single-nucleotide variants may represent distinct, closely related taxa, we caution against this overly simplistic interpretation due to the fact that many bacterial genomes contain multiple polymorphic copies of the 16S gene[12–14]. We performed PacBio CCS sequencing of a 36 species bacterial mock community (Supplementary Table 3 and Supplementary Fig. 4) to demonstrate (i) that the 16S sequence of many bacteria varies between operons within the

same genome and (ii) that high-throughput sequencing is sufficiently accurate to resolve these intragenomic differences.

We aligned PacBio full-length 16S sequences to a reference database containing a single representative 16S sequence for each member of our mock community and used the alignment statistics to evaluate the accuracy of this sequencing approach. Comparing the number of passes used to generate a CCS with the occurrence of single-nucleotide substitutions, insertions and deletions indicated that ten passes could minimize these combined errors to a minimum frequency of < 1.0% (although it was notable that the minimum achievable error varied between sequencing runs; Supplementary Fig. 5). However, we did observe a coincidence of deletion errors with the location homopolymer runs in our reference sequences (Supplementary Fig. 6), which was not nucleotide-specific and was exacerbated by the length of the sequenced homopolymer (Supplementary Fig. 7). We subsequently validated deletions within the *Escherichia coli* 16S gene using Illumina whole genome shotgun (WGS) sequencing, which demonstrated that only one of the deletions occurring in PacBio sequences was genuine (Supplementary Fig. 8).

Satisfied that CCS sequencing can produce 16S reads with a low frequency of substitution errors, we next reasoned that a proportion of the substitution errors within accurately aligned reads should reflect variation attributable to 16S polymorphisms within a species' genome[12]. For example, reads aligned to the *E. coli* strain K-12 substr. MG1655 showed a substitution profile, which mirrored exactly that predicted by aligning all seven of the 16S sequences known to be present in this genome[15] (Fig. 2a, c). We were further able to validate the stoichiometry of these nucleotide substitutions by quantifying variation in comparably aligned Illumina WGS reads (Fig. 2b) and demonstrate that a similar substitution profile was reproducible across multiple

sequencing runs (Supplementary Fig. 9). Alignments to other reference sequences in our mock community showed a similar trend of abundant substitutions localized to specific base positions along the 16S gene, although we note that the signal-to-noise ratio increased significantly when the 16S gene in question had fewer than 100 aligned reads (Supplementary Fig. 10).

The observation that long-read sequencing can identify 16S polymorphisms within the same genome has important implications. First, it demonstrates that it is not valid to assume that high-throughput sequence reads differing by one or few nucleotides represent a distinct taxa[6,16]. Within a single genome, two or more 16S sequences may be identical, whereas others may be unique. Correspondingly, some homologous 16S loci may retain identical sequence between two closely related strains, whereas others may have diverged at one or few nucleotide positions. In this context, any community-level or taxonomic interpretation of 16S data should ideally account for the fact that the relative abundance of 16S sequences arising from very closely related taxa will reflect a linear combination of (i) the frequency with which each unique sequence is represented across genomes and (ii) the relative abundance of the genomes for each taxon.

Second, although intragenomic 16S sequence variation complicates community-level analysis, it also has the potential to increase the power of the 16S gene to discriminate between closely related taxa, because it enables sequence-based comparison to extend across multiple divergent loci. For example, sufficient nucleotide variation exists to distinguish *E. coli* strain K-12 MG1655 from the enterohemorrhagic strain O157 Sakai (Fig. 2c, d). Thus, we argue that, when appropriately accounted for, multiple polymorphic 16S copies are not an inconvenience to be overlooked, rather they will enable the 16S gene to be used in

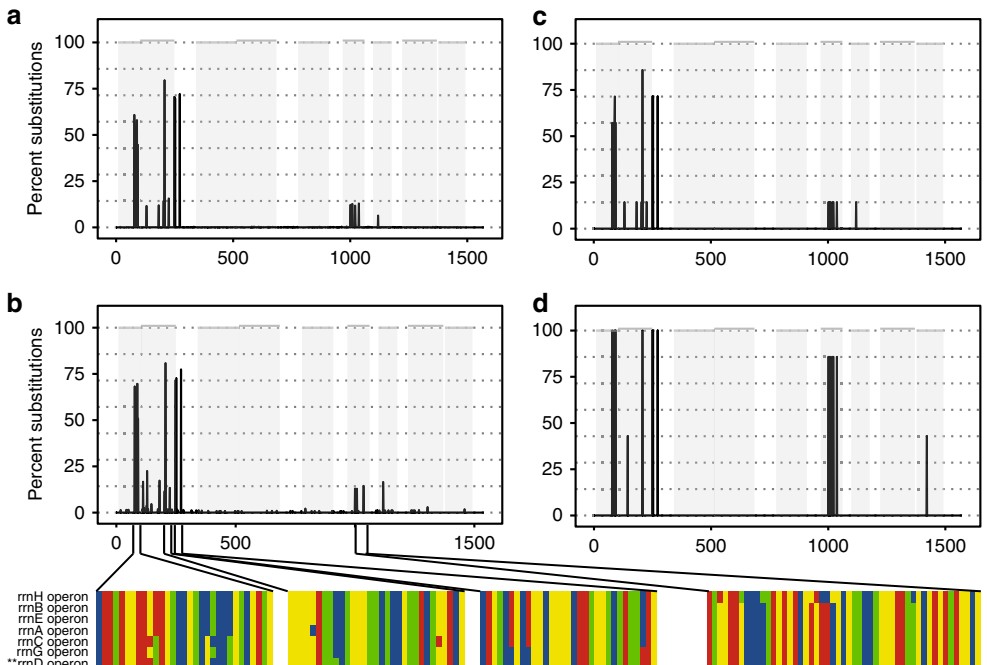

**Fig. 2** Polymorphisms in *E. coli* 16S rRNA gene sequences. **a** The position and frequency of substitutions appearing in *E. coli* strain K-12 MG1655 V1–V9 amplicons generated from our mock community and sequenced on the PacBio RS II platform. **b** The position and frequency of substitutions in reads generated from genomic sequencing of the isolated *E. coli* strain K-12 MG1655 on the Illumina MiSeq platform. Magnified regions show respective positions in the alignment of all seven 16S genes present in the *E. coli* K-12 MG1655 reference genome. The 16S sequence from the rrnD operon (**) is used as the reference for all SNP phasing. **c** The predicted nucleotide substitution profile of *E. coli* K-12 MG1655 based on aligning the seven 16S gene sequences present in the reference genome. **d** The predicted substitution profile of *E. coli* O157 Sakai based on aligning the seven 16S gene sequences present in the reference genome. Gray panels depict variable regions defined by commonly used primer-binding sites (Supplementary Table 1). Dashed lines indicate the expected proportion of nucleotide substitutions, given there are seven 16S gene copies within each genome. Source data are provided as a Source Data file

strain-level microbiome analysis. We also note that the power of intragenomic 16S sequence variation to discriminate closely related taxa is likely to diminish when partial 16S sequences are used. For example, SNPs distinguishing the *E. coli* strains K-12 MG1655 (Fig. 2c) from O157 Sakai (Fig. 2d) are found in variable regions V1, V2, V6, and V9.

**16S polymorphisms can be resolved in vivo**. Microbiome communities are often complex, existing in diverse biochemical environments (e.g., stool, saliva, sputum, etc.) and containing many hundreds of unique taxa whose relative abundance spans a broad dynamic range. This complexity is not well represented in either in-silico or mock community experiments. We therefore performed an additional experiment to demonstrate that sequencing of the full 16S gene while accounting for intragenomic 16S SNPs can resolve closely related bacterial taxa in vivo.

We carried out PacBio CCS sequencing of the V1–V9 region for four human stool samples collected from healthy adult volunteers. For comparison, we sequenced the V1–V3 region using the Illumina MiSeq and, to provide a benchmark for species-level taxonomic quantification, we performed metagenomic WGS (mWGS) sequencing using the Illumina NextSeq. To evaluate the extent to which each of these sequencing approaches can resolve closely related taxa, we focused on the genus *Bacteroides*. In addition to being abundant in the human gut, this genus is highly diverse, containing multiple species that can exert both good and bad effects on human health[17]. It has also been used previously as a model taxon for demonstrating the utility of the 16S gene for high-resolution taxonomic analysis[18].

When we calculated *Bacteroides* abundance at the genus level, V1–V9 sequencing and V1–V3 sequencing produced comparable results. Both approaches identified two individuals with low *Bacteroides* relative abundance (~10–25%) and two individuals with high *Bacteroides* relative abundance (~40–60%; Fig. 3a). However, species-level quantification via mWGS sequencing revealed far greater diversity, with a different *Bacteroides* species dominant in the gut of each individual (Fig. 3b and Supplementary Data 1). When clustering OTUs at 99% identity, both V1–V9 and V1–V3 sequencing were able to reflect this species-level variation (Fig. 3b), with the notable exception that V1–V3 sequencing did not detect *Bacteroides intestinalis*, which was abundant in one of the four human gut microbiome samples. Based on these results we conclude that, when used in conjunction with an appropriate identity threshold (e.g., 99%), OTU-based approaches have the potential to resolve species-level diversity observed in the human gut. We further note that, although full-length 16S sequencing may be optimal for species-level analysis, highly informative variable regions (e.g., V1–V3) may also be adequate for this purpose.

Taking advantage of the fact that *Bacteroides vulgatus* was present at high relative abundance in two of our human gut microbiome samples, we next asked whether intragenomic variation between 16S gene copies could be detected in vivo. We aligned every full-length sequence classified as belonging to our *B. vulgatus* V1–V9 OTUs (Fig. 3b and Supplementary Data 1) to a single representative *B. vulgatus* 16S gene sequence. We then compared the resulting nucleotide substitution profiles (Fig. 3c) with profiles predicted from two reference genomes present in the NCBI RefSeq database[19] (Fig. 3d).

The majority of nucleotide variation present in our in vivo generated *B. vulgatus* OTU reflected true variation attributable to intragenomic polymorphisms. In contrast, variation likely due to sequencing errors appeared low and well below the minimal ~14% frequency that would be expected if there were a single *B. vulgatus* strain in each sample with seven 16S gene copies in its genome (Fig. 3c, dashed lines).

Although we did not know the true number of *B. vulgatus* strains present in each in-vivo sample, it was notable that both nucleotide substitution profiles bore closer resemblance to strain ATCC 8482 than mpk. Variation also existed at specific loci that could potentially indicate meaningful differences between the in vivo and ATCC 8482 reference genomes. For example, a single polymorphism was detected in the V5 region of ATCC 8482, which was present in three 16S copies (43%). In the first in-vivo sample (Scott) this polymorphism was present in 84% of reads, whereas in the second (IronHorse) it was present in 69% of reads. These numbers correspond closely to the numbers expected if a polymorphism were present six and five out of seven 16S genes, respectively.

In conclusion, we show that full-length 16S sequencing of the human gut microbiome can accurately resolve single-nucleotide substitutions that reflect intragenomic variation between 16S gene copies. The presence of such variation indicates that 16S sequences must be clustered to reflect meaningful taxonomic units. Using OTUs clustered at 99% identity, we show that full-length 16S has the potential to provide species and even strain-level taxonomic resolution. Analysis of microbial communities at these taxonomic levels promises to provide a very different perspective to the one afforded by genus-level abundance estimates.

**Intragenomic 16S polymorphisms are highly prevalent**. Having demonstrated that it is possible to resolve intragenomic copy variants in vivo, we next sought to establish the extent to which such copy variants appear in taxa commonly found within the human gut microbiome. We further sought to establish whether such profiles can routinely be used to distinguish between strains of the same species.

We cultured 381 taxa from the gut microbiome of the healthy individuals depicted in Fig. 3, as well as from other individuals participating in the same original study[20] (Supplementary Data 2). We subsequently performed full-length 16S gene sequencing on isolates and aligned sequenced reads to identify nucleotide substitutions characteristic of intragenomic 16S gene copy variants.

Taxonomic classification of isolates identified 58 putative species (Supplementary Data 2), while clustering a single representative sequence for each isolate at 99% similarity resulted in 61 OTUs (with between 1 and 73 isolates assigned to each OTU). In total, 349 of 381 sequenced isolates (54 of 61 OTUs) had one or more SNP, indicating the presence of 16S gene polymorphisms, and 205 unique SNP profiles were identified when accounting for potential sequencing error (Fig. 4a and Supplementary Data 2).

Notably, comparing SNP profiles for isolates assigned to the same OTU frequently revealed differences in the frequency of SNPs that were suggestive of differences in intragenomic 16S gene copies between closely related taxa. Examples of different substitution profiles are shown for three taxa (Fig. 4b–d), which are suggestive of strain-level variation comparable to that we demonstrated in principle for *E. coli* (Fig. 2b).

In conclusion, we show that many of the culturable members of the human gut microbiome frequently possess 16S gene polymorphisms, which, when properly accounted for, have the potential to resolve strains of the same species.

## Discussion

Here, we have presented the results of four experiments that collectively demonstrate the taxonomic resolution achievable in the current 16S gene-based microbiome studies. In particular, we have focused on whether sequencing the full 16S gene while

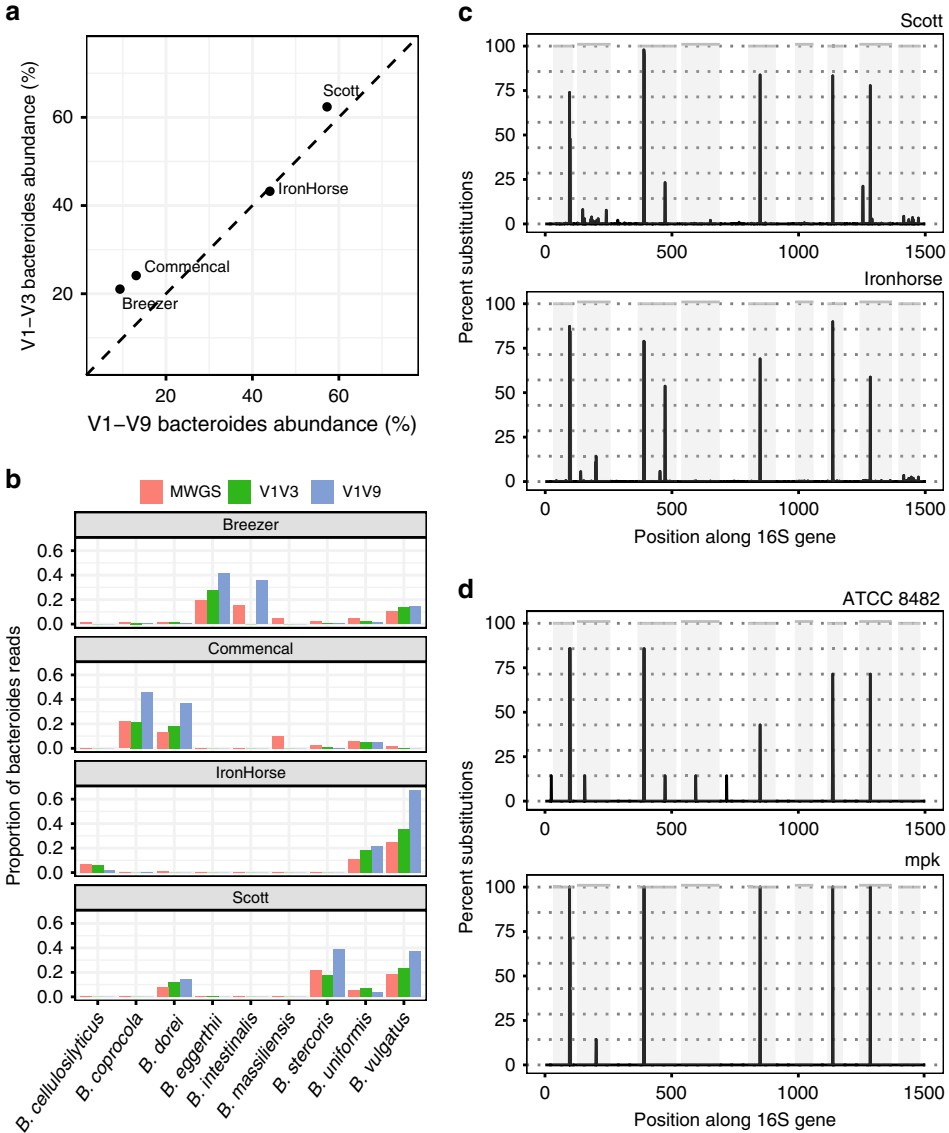

**Fig. 3** Detecting *Bacteroides* in human stool samples. **a** The relative abundance of the genus *Bacteroides* in four human stool samples quantified using either V1–V9 amplicons (*x*-axis) or V1–V3 amplicons (*y*-axis). **b** The relative abundance of *Bacteroides* species in the same four samples. Species abundance was quantified from mWGS sequencing or from V1–V3/V1–V9 OTUs generated at 99% identity. Abundance is shown for the most abundant species as quantified by mWGS (for abundance estimates of all *Bacteroides* species detected by each platform, see Supplementary Table 5). **c** Nucleotide substitution profiles generated by aligning all V1–V9 amplicon sequences assigned to the single OTU identified as *Bacteroides vulgatus*. Profiles are shown for the two stool samples with high *B. vulgatus* relative abundance (IronHorse and Scott). **d** Nucleotide substitution profiles predicted from the reference genomes of two different *B. vulgatus* strains ATCC 8482[39] and mpk[40]. In both **c** and **d**, nucleotide substitutions were identified relative to a single reference 16S gene for *B. vulgatus* ATCC 8482 (NCBI Gene ID 5304800). Gray panels depict variable regions defined by commonly used primer-binding sites (Supplementary Table 1). Dashed lines indicate the expected proportion of nucleotide substitutions, given there are seven 16S gene copies within each genome. Source data are provided as a Source Data file

accounting for 16S gene copy variants makes the detection of bacterial species and strains a realistic prospect.

High-throughput sequencing of the full 16S gene with sufficient accuracy to discriminate between copy variants has until recently been constrained by a lack of available sequencing technologies. The advent of long-read approaches on Nanopore[4] and PacBio[3] platforms has changed this. Several previous studies have provided detailed evaluation of PacBio CCS for targeted amplicon sequencing[21–24], and some have demonstrated this approach is capable of improving discrimination between bacterial species present in microbial communities[24,25].

Although our study necessarily addresses important technical details, its goal is to explore the full potential of the 16S gene for

discriminating bacterial taxa rather than re-evaluate a particular sequencing technology. In addressing this goal, however, we highlighted the prevalence of sequencing errors in PacBio CCS reads as a factor that limits the ability to resolve highly similar sequences. A particular problem was deletion errors coincident with homopolymer runs in the target sequence. Although random sequencing errors may be overcome by increased sequencing depths, such systematic errors may occur at a given frequency and hence may not be improved by greater sequencing effort. Future work would benefit from explicitly determining how recent advances in sequencing platforms, chemistries, and computational approaches can improve these errors.

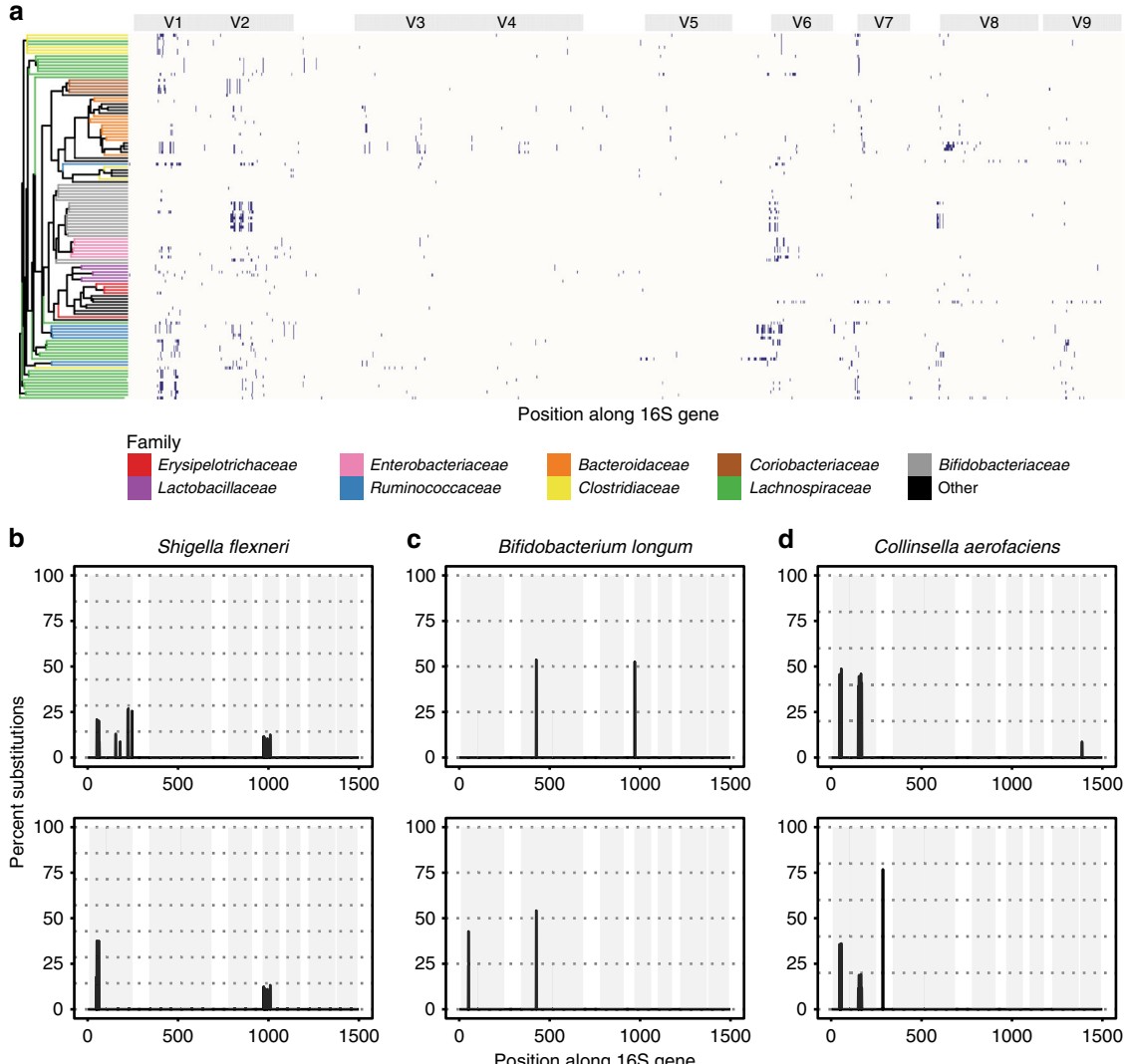

**Fig. 4** Intragenomic 16S gene polymorphisms in human gut microbiome isolates. **a** Location of SNPs present in the 16S genes of individually cultured bacterial isolates. SNP locations were identified through phasing full-length 16S gene sequences generated for each individual isolate. *X*-axis denotes position along the 16S gene. *Y*-axis denotes individual isolates clustered based on their inferred phylogeny. Dark blue region indicates the location of a polymorphism. For clarity, a maximum of five isolates belonging to the same species are shown. For details of nucleotide substitution profiles for all sequenced isolates, see Supplementary Data 2. **b–d** Examples of nucleotide substitution profiles showing strain-level differences between isolates identified as belonging to three bacterial species: **b** *Shigella flexneri*; **c** *Bifidobacterium longum*; **d** *Collinsella aerofaciens*. For each species, two isolate nucleotide substitution profiles are shown; however, additional examples can be found in Supplementary Data 2. Isolates were identified as belonging to the same species if their representative sequences were assigned to the same OTU when clustering at 99% sequence identity. Taxonomic identification was performed using BLAST to align representative sequences to the NCBI 16S BLAST database (see Methods). Gray panels depict variable regions defined by commonly used primer-binding sites (Supplementary Table 1). Dashed lines indicate the expected proportion of nucleotide substitutions, given the number of 16S gene copies predicted for each genome. Source data are provided as a Source Data file

Given the emphasis of our study, we chose to overcome such platform-specific errors by focusing on substitutions and ignoring the contribution of insertions and deletions to intragenomic 16S gene copy polymorphisms. The presence of a single deletion in one of the seven *E. coli* strain K-12 substr. MG1655 16S genes demonstrates that this is an imperfect approach. However, we argue that the contribution of insertions/deletions to intragenomic polymorphisms is likely to be small relative to the contribution of substitutions[12]. Therefore, current limitations that may be specific to the sequencing approach used do not invalidate our investigation of full-length 16S gene sequencing as a viable method for discriminating between species and strains. In the ensuing discussion, we address several important conclusions from this investigation.

First, we conclude that sequencing the entire 16S gene provides real and significant advantages over sequencing commonly targeted variable regions. Although the 16S gene will never provide a perfect representation of bacterial species diversity[26], none of the variable regions covered by partial 16S sequencing were able to recapture the diversity represented when sequencing the full ~1500 bp gene. Assuming our in-silico experimental dataset provided a reasonable approximation of bacterial species, we conclude that most variable regions are sufficient to identify genera, but they are unlikely to ever adequately discriminate between species. In consequence, irrespective of the resolution at which they are clustered, variable regions will likely underrepresent the true species richness of a microbiome sample.

Second, we argue that intragenomic variation in the 16S gene should not be ignored. In particular, we caution against the conclusion that quantifying exact sequence variants (ESVs) is preferential to more traditional OTU-based approaches[16]. This conclusion assumes that ESVs represent a more meaningful taxonomic unit than OTUs. Given that the majority of bacterial isolates we sequenced contained multiple, variant copies of the 16S gene within their genome, this assumption may not always be correct. The potential for 16S copy variants to bias estimates of bacterial diversity is well established[27], and we and others[25] have shown the number of unique sequences detected in a mock community is far greater than the number of species known to be present.

We note that, similar to OTUs, ESVs do not need to accurately represent individual taxa, to be useful and informative[18]. However, our results show that quantifying ESVs will likely overestimate species richness, just as OTUs based on variable regions may underestimate it. As a means of quantifying individual taxa, ESVs may also be limited, due to the fact that multiple unique sequences originating from the same genome are not necessarily present at the same relative abundance (e.g., E. coli has seven 16S gene copies, of which six are unique for strain K-12 MG1655, whereas four are unique for O157 Sakai). Although these caveats do not preclude the use of ESVs as useful indicators of either taxonomy or diversity, they must necessarily be accounted for when interpreting results.

Third, we argue that appropriate clustering of intragenomic 16S gene sequence variation can in fact be a valuable method by which to provide accurate representation of bacterial species. Previous studies have reported intragenomic 16S gene polymorphisms as a problem that potentially confounds bacterial species richness estimates[25,27]. By contrast, we demonstrate that, when handled correctly, the presence of such polymorphisms in full-length 16S reads has the potential to aid in taxonomic classification. Our in-vivo experiment demonstrated that full-length 16S gene sequences clustered at 99% identity provided reasonable estimates of Bacteroides species relative abundance when compared with mWGS-based quantification. More importantly, these 99% OTUs also appeared to adequately cluster the seven 16S gene copies present in the B. vulgatus genome. Although we stopped short of determining how well the 99% identity threshold separates intragenomic vs. inter-genomic sequence variation for all bacterial species, we note that other studies have previously endorsed similar thresholds for full-length 16S studies[26,28].

Finally, by extensive culturing of bacteria present in the human gut microbiome, we provide support for the observation that intragenomic 16S gene copy variants are present in a significant proportion of bacterial taxa[12,27]. Phasing of 16S gene SNPs produced highly similar substitution profiles for closely related taxa, indicating that these profiles provide a robust method for species-level taxonomic identification. Furthermore, assuming that 99% sequence similarity is an adequate threshold for clustering sequences originating from the same genome, differences in SNP profiles, reflecting polymorphisms in one or more 16S gene copies reproducibly reflected differences between strains of the same species.

In conclusion, our results demonstrate that appropriate handling of high-throughput, full-length 16S sequence data has the potential to enable accurate classification of individual organisms at very high taxonomic resolution.

## Methods

**In-silico comparison of full vs. partial 16S gene sequencing**. The in-silico analysis was carried out separately on two non-redundant public databases: Greengenes v13.8.99[29] and the Human Oral Microbiome Database (HOMD) v13[30]. Only the results for the Greengenes database are reported in the main text. For the HOMD, a single sequence was randomly selected to represent each species present in the database. As Greengenes does not consistently provide species-level taxonomic classification, all sequences with genus-level classification were selected and sequences representative of 99% sequence-similarity clusters were used to represent distinct species. Supplementary Fig. 2a (and Source Data) indicate the relative extent to which different bacterial taxa were represented within this Greengenes-derived database.

In-silico amplicons demarcating different sub-regions of the 16S gene were generated by trimming regions defined by established primer sets (Supplementary Table 1) using Cutadapt v1.4.2[31], allowing up to three mismatches within the primer alignment. Sequences were discarded if one or more variable region (including V1–V9) could not be identified by the trimming tool, contained N's, or if the resulting amplicon was >2 SDs away from the observed mean length for the respective region. These curation steps retained 15% and 75% of the sequences in the Greengenes and HOMD databases, respectively (Supplementary Table 2). Full-length (V1–V9) amplicons were aligned using MUSCLE[32] and Shannon entropy was calculated at each base position along a single E. coli str. K-12 substr. MG1655 (Fig. 1a) 16S gene sequence (NCBI Gene ID 947777). Accordingly, deletions within other 16S sequences are represented in entropy plots, whereas deletions within the reference sequence are not.

To determine the taxonomic resolution of afforded by different variable regions, each in-silico amplicon was classified against the filtered reference database from which it was generated using the mothur command classify.seqs[33] with a range of minimum confidence thresholds (-cutoff 30–98). To create OTUs, in-silico amplicon datasets generated for each sub-region were filtered to remove non-unique sequences and re-ordered to correspond with the sequence order in the V1–V9 dataset. Each amplicon was assigned a unitary abundance and OTUs were generated at a variety of similarity thresholds (97%, 98%, and 99%) using the USEARCH command cluster_otus[34], with chimera detection disabled using the option -uparse_break −999.

**Construction of a bacterial mock community**. Based on data available from the Human Microbiome Project and Human Oral Microbiome database, 36 bacterial strains were selected to represent microbes prevalent in the human body sites including the airways, gut, oral cavity, skin, and vaginal tract (Supplementary Table 3). DNA from ten strains was obtained directly from ATCC (www.atcc.org). The other 26 strains were cultured in appropriate media and environmental conditions until cultures reached late logarithmic phase (Supplementary Table 3)[35–38]. Unless otherwise indicated, anaerobes were grown under an atmosphere of 90% $N_2$, 5% $H_2$, and 5% $CO_2$. DNA was isolated by suspending cultures in TE buffer containing 20 mg ml$^{-1}$ lysozyme and incubated at 37 °C for 30 min. Subsequently, AL buffer (Qiagen, Valencia, CA) containing 1.23 mg ml$^{-1}$ Proteinase K was added and samples were incubated at 56 °C overnight. Samples were then incubated at 95 °C for 5 min and DNA was isolated using a DNeasy Blood and Tissue kit (Qiagen). DNA was eluted in MD5 solution (MoBio Laboratories, Carlsbad, CA). Isolated DNA was pooled in a manner that accounted for different numbers of 16S rRNA gene copies per species. Briefly, the genome size ($n$) in bp was estimated for each organism and was used to calculate the mass of DNA ($m$) per genome using the formula $m = (n) (1.096 \times 10{-}21\,\text{g bp}^{-1})$. Genome mass was then normalized based on the predicted copy number of the 16S rRNA gene (Supplementary Table 3) and the appropriate mass of DNA containing the required 16S copy number for each species was calculated.

**Illumina library preparation shotgun sequencing and assembly**. WGS sequencing was performed for 19 members of the mock community that did not have WGS sequence data publicly available. Libraries were made using the Illumina TruSeq Nano DNA HT kit according to the manufacturer's instructions, and were sequenced on either the Illumina MiSeq or HiSeq platform. Genomes for sequenced organisms were assembled individually using SPAdes v3.5.0[39] with post-processing enabled (–careful).

**PacBio library preparation and sequencing**. Sequencing libraries were prepared by amplifying the V1–V9 region of the 16S rRNA gene using primers 27F and 1492R (Supplementary Table 1), and Accuprime Taq polymerase (Thermo Fisher Scientific, Waltham, MA). Amplicons were purified using PCR purification kits (Qiagen, Hilden, Germany) and 1 µg of DNA was used for the SMRTbell 1.0 Template Prep Kit (Pacific Biosciences, Menlo Park, CA). SMRTbell-adapted sequences were run on the Pacific Biosciences (PacBio) RS II platform using P6C4v2 chemistry. Output files were processed and assembled into CCS reads using CCS2 v3.0.1 setting the minimum passes to 3, minimum signal-to-noise ratio (SNR) to 4, minimum length to 1200, minimum predicted accuracy to 0.9, and the minimum $Z$-score to −5. Consensus sequences longer than 1600 bp were discarded.

**Analysis of the bacterial mock community**. Reference 16S rRNA gene sequences matching strains in the mock community were initially downloaded from the RDP database[40]. Several reference gene sequences contained ambiguous base calls. Each sequence was therefore aligned to its respective WGS assembly and the aligned assembly region extracted to create an improved reference gene set containing a

single representative 16S rRNA gene sequence for each member of the mock community.

To determine sequence variation in PacBio CCS data, reads generated from the mock community were aligned to the mock reference gene set using Cross_match[41] with the minimum alignment score (-minscore) set to 750, the substitution penalty (-penalty) set to −9, and only the best alignment for each read reported (-masklevel 0). Output alignments were parsed to determine the number and location of insertions, deletions, and substitutions in reads aligning to each reference 16S rRNA gene sequence.

To determine the frequency and position of expected sequence variation—attributable to the presence of multiple, divergent copies of the 16S rRNA gene within a single genome—the seven gene copy variants known to exist in the E. coli K-12, MG1655 sub-strain (NC_000913.3) were downloaded from RefSeq and aligned using MUSCLE. To provide a second estimation of expected intra-genome sequence variation, Illumina WGS sequence reads were aligned to the single E. coli reference sequence present in the mock community reference database and the location of insertions, deletions, and substitutions inferred using the SAMtools pileup command[42].

**Sampling and sequencing of the human stool microbiome.** Stool samples were collected from four healthy, competitive cyclists enrolled in the study described by Petersen et al.[20] Informed consent was obtained from all human participants and work was carried out with the oversight of the Jackson Laboratory Internal Review Board (IRB numbers 1503000013 and 16-JGM-07). Fecal material was self-collected using polyethylene sample collection containers (Fisher Scientific) and was placed on freezer packs before shipping to the Jackson Laboratory for Genomic Medicine. Once received, samples were stored at −80 °C prior to extraction. DNA was extracted using the PowerSoil DNA Isolation Kit (MO BIO Laboratories, Inc.). mWGS sequence libraries were prepared as described for the bacterial mock community and 150-base paired-end reads were generated on the Illumina Next-Seq platform. Exact duplicate sequences were discarded on the assumption that they were PCR artifacts and the remaining reads were screened against the human reference genome (GRCh38) using BMTagger[43]. Adapters and low-quality bases were trimmed using Flexbar[44].

Amplicon libraries were prepared and sequenced for the V1–V9 region (PacBio RS II) and V1–V3 region (Illumina MiSeq) as described for the bacterial mock community.

**Quantifying bacteroides in the human stool microbiome.** Taxonomic abundance estimates were generated from mWGS data by aligning sequenced reads to the Real Time Genomics™ (RTG) reference database of bacterial genome assemblies (v2.0), using the map and species commands within the RTG-core bioinformatics package (www.realtimegenomics.com/products/rtg-core).

Amplicon sequence data for the V1–V3 and V1–V9 region of the 16S rRNA gene were pooled and de-replicated using USEARCH (v8.0.1517), before being clustered into OTUs at either 97% or 99% similarity thresholds using the -cluster_otus command[34]. Amplicon sequences from each sample were then reassigned to each OTU at the same similarity threshold used for clustering in order to obtain OTU relative abundance estimates. The genus of each OTU was determined using the RDP classifier v2.2[11] in conjunction with the Greengenes database, v13.5 at a confidence threshold of 0.8.

V1–V3 and V1–V9 amplicons belonging to the genus *Bacteroides* were selected by directly classifying individual amplicon sequences using the RDP classifier. Sequences were then clustered into OTUs at either 97% or 99% identity thresholds using USEARCH. Representative sequences of *Bacteroides* OTUs generated for each variable region/identity threshold combination were assigned a putative species classification by aligning each sequence to the RTG reference database (v2.0) using the USEARCH local alignment algorithm[45], allowing up to 50 top hits for each aligned sequence.

The suitability of the RTG database as a reference for discriminating different *Bacteroides* species was assessed by extracting the 16S rRNA gene sequences for each *Bacteroides* genome contained therein. Extracted sequences were globally aligned using MUSCLE, a maximum-likelihood tree was constructed using FastTree v2[46], and visualized using the R package ape[47]. The resulting tree (Supplementary Fig. 11) indicated that sequence variation within the 16S gene was sufficient to resolve most major *Bacteroides* species contained within this database.

The suitability of either 97% or 99% identity thresholds for clustering V1–V3 and V1–V9 amplicons at the species level was assessed by determining the frequency with which OTUs for each variable region/identity threshold aligned optimally to a single species in the RTG reference database (Supplementary Fig. 12).

V1–V9 amplicon sequences assigned to the single OTU identified as *B. vulgatus* (OTU_1; Supplementary Data 1) were detected at high relative abundance in two human stool microbiome samples (Scott and IronHorse). Sequences from each sample were therefore extracted and aligned to the single 16S rRNA gene reference sequence used in the mock community analysis. Sequence alignment was performed using Cross_match and alignment errors were calculated as described above.

**Isolation and sequencing of bacteria from human stool.** Stool samples were again contributed by competitive cyclists enrolled in the study described by Petersen et al.[20] Ethical oversight and sample collection were as described above. Bacteria were cultured on a variety of media and under anaerobic conditions, unless otherwise stated (Supplementary Data 2). Individual colonies were picked and DNA extracted using the MasterPure™ Gram Positive DNA Purification Kit (Lucigen). Samples were multiplexed and sequenced on a PacBio RS II. A subset of multiplexed libraries were sequenced on multiple SMRT cells at varying loading concentrations (Supplementary Data 2) resulting in different numbers of total reads. Each repeated run was therefore treated as a technical replicate to determine (i) the measurement error for the estimation of intragenomic 16S gene SNP frequencies attributable to the sequencing platform and (ii) the relationship between measurement error and sequencing depth.

**Computational analysis of individual isolates.** Sequence data for each isolate were quality filtered and adapters removed as described above. Filtered sequences were reoriented using the mothur command align.seqs, with the Silva gold database as a reference and the arguments flip = t, threshold = 0.5. Gaps in alignments were subsequently removed with the mothur command degap.seqs. Filtered, reoriented fasta files were then de-replicated using the USEARCH command -derep_full-length and then sorted with -sortbysize, with the argument -minsize 1. The most abundant unique sequence for each isolate was then extracted (on the assumption it was the least likely to contain sequencing errors) and was used as a reference against which to align all reads for that isolate. Sequence alignment was performed using Cross_match with the arguments -minscore 1200, -masklevel 0, and alignment errors (substitutions, insertions, and deletions) calculated as described above.

Due to the prevalence of sequencing errors in processed reads (e.g., Supplementary Fig. 10), insertion and deletion errors were ignored when generating nucleotide substitution profiles. Substitution errors in alignments were filtered in a multi-step process to separate true intragenomic SNPs from background error. First, samples with fewer than 200 aligned reads were discarded, because preliminary investigation indicated they had insufficient signal-to-noise ratio for the detection of true SNPs. Second, the distribution of the frequency of substitution errors was calculated across the entire aligned region of the 16S gene. Base positions where the substitution error frequency was well outside instrument error (nine interquartile ranges above the upper quartile) were identified as true SNPs. Finally, samples with SNPs at >3% of base positions were discarded, as this threshold was empirically determined to exclude impure isolates.

We assessed SNP measurement error ($\zeta_w$)[48] for a subset of cultured isolates where replicate sequencing was performed on multiple SMRT cells using varying input library concentrations (Supplementary Data 2). We also took advantage of variation in sequencing depth between replicates to determine whether the measurement error was affected by the number of reads available for SNP phasing. Across 271 samples, the median $\zeta_w$ was 1.8% (Supplementary Fig. 13a). There was no obvious relationship between measurement error and sequencing depth for samples with > 200 reads (Supplementary Fig. 13b).

**Taxonomic identification of sequenced isolates.** Isolates were assigned a putative taxonomy using BLAST[49]. The most abundant unique sequence for each isolate was searched against the NCBI 16S Microbial database using blastn, with the argument -max_target_seqs 20. Resulting hits were sorted first by e-value, then bitscore and the taxonomy of the highest scoring sequence was reported. In addition, sequences were clustered into OTUs at 99% sequence identity using USEARCH command -cluster_otus with the arguments -otu_radius_pct 1.0, -uparse_break −999. The phylogenetic relationship between isolates was determined by aligning the most abundant unique sequence for each isolate, then constructing a maximum-likelihood tree using FastTree v2.

To determine the total number of unique nucleotide substitution profiles generated from sequenced isolates, all isolates identified as belonging to the same OTU were compared with one another. Two isolates were considered different if the substitution frequency at one or more SNP loci differed more than 3 SDs above the mean measurement error (i.e., 6.58%, Supplementary Fig. 13).

**Reporting summary.** Further information on research design is available in the Nature Research Reporting Summary linked to this article.

## Data availability

Sequence data that support the findings of this study are available via the NIH Sequence Read Archive. Sequence data for the mock community are available via BioProject PRJNA552603. mWGS and V1–V3 amplicon sequence data for human microbiome samples that were published previously by Petersen et al.[20] are available via BioProject PRJNA305507 (Breezer V1–V3: SRX147975, Scott V1–V3: SRX1479742, IronHorse V1–V3: SRX1479743, Commencal V1–V3: SRX1479751, Breezer mWGS: SRX1479791–5, Scott mWGS: SRX1479846–50, IronHorse mWGS: SRX1479811–5, Commencal mWGS: SRX1479787). V1–V9 amplicon sequence data for human microbiome samples are available via BioProject PRJNA552603. V1–V9 amplicon sequence data for bacterial isolates are available via BioProject PRJNA561528. Data

underlying Figs. 1–4 and Supplementary Figs. 1–13 are provided as Source Data. All other data are available from the corresponding author upon reasonable request.

## Code availability

A copy of the code used for the analyses reported in this manuscript can be found at: https://github.com/TheJacksonLaboratory/weinstock_full_length_16s.

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

## Acknowledgements

This work was supported by an NIH Common Fund Human Microbiome Project grant (1U54DE02378901) and National Institutes of Health, National Institute of Diabetes and Digestive and Kidney Diseases grant (R01DK088831) awarded to G.M.W. We thank Sarah Kremer, Chaman Ranjit, and Kathleen Teter for help with culturing isolates.

## Author contributions

B.Y.H. created bacterial mock community. L.M.P. collected human gut microbiome samples and supervised strain collection and sequencing. J.S.J., D.J.S., P.D., L.C., E.S., S.L., B.M.H., H.O.A., B.Y.H., and G.M.W. contributed to data analysis. G.M.W. designed study. G.M.W. and M.G. oversaw the study. J.S.J., D.J.S., and G.M.W. wrote the paper.

## Competing interests

The authors declare no conflict of interest.

## Additional information

**Peer Review Information** *Nature Communications* thanks the anonymous reviewers for their contribution to the peer review of this work. Peer reviewer reports are available.

