## [Peer Review File · Nature Communications]

Reviewers' comments:

Reviewer #1 (Remarks to the Author):

This manuscript by Johnson et al is a well written and well presented; it represents a substantial amount of computational work along with somewhat less in the way of original laboratory-based studies to compare (in silico) long and short-read 16S microbiome data for determining species- (and strain-) level composition. The authors correctly concluded that by sequencing the entire 16S gene using PacBio's circular consensus sequencing protocol (as opposed to Illumina-based short-read partial 16S gene sequencing) that it is possible in a far higher percentage of cases to obtain species- and in some cases strain-specific data. They authors also correctly point out that it is important to be aware of the fact that individual bacteria have multiple copies of the 16S operon, and that therefore counting each high quality sequence with a different SNP profile as a different strain (species) could inflate the complexity of the reported microbiome data.

Whereas these data and their analyses are state-of-the-art, and the results and interpretation are highly believable - they are essentially derivative of several recent papers in the microbiome space - particularly Earl et al Microbiome 2018 (which they reference). In Earl et al, not only do they reach essentially the same conclusions, but they also go much further and develop a full length 16S database (which is available to the public) for species-specific ID. Moreover, Earl et al analyze multiple (as opposed to a single) mock microbiomes - including one that is approximately 10 times the complexity (in terms of species numbers) that the current paper does - and they perform detailed analyses on a large set of novel specimens from the human nasopharynx (something not undertaken in the current manuscript).

Importantly, in the Earl et al paper they also compare, in silico, the resolving power of short and long read sequences with regard to species-specificity and reach essentially the same conclusions as that of the authors of the current paper. Thus, although this reviewer has no quarrel with the results presented, there is little that is truly novel in this paper, excepting perhaps their dwelling on the multiple 16S operon phenomenon (which was addressed in Earl et al in the construction of the full-length 16S database).

Reviewer #2 (Remarks to the Author):

Johnson et al. present in silico and sequencing data to support the prospect of full 16S gene sequencing using single molecule real-time sequencing methods of the 1500 bp region. Using these methods, the authors demonstrate strain level resolution of bacterial communities, both experimentally mixed and primary. Also, the authors spend considerable effort to show CCS accuracy of long reads enables accuracy to properly phase SNP level discordances between intra-genomic copies of 16S in a metagenomic system. This approach further underscores the need for intragenomic analyses approaches not achievable with current short read, targeted 16S approaches that only leverage specific variable regions for taxonomic analyses of complex samples today.

The molecular methods (primer design and sequencing) are all now fairly common methods from publications in the 2016-2018 era enabling full-length 16S sequencing using SMRT-Seq and have been applied broadly to various metagenomics communities since – to show similar findings. Where this paper is novel is in the specific focus on the intragenomic profiling of variation between 16S copies, which is often overlooked by other publications. The first major claim for the paper is more follow up to recapitulate the utility of full-length 16S sequencing relative to papers like Wagner et al, 2016 (BMC uBiology), whereas the second claims on intragenomic and strain level demonstrations are much more novel and of interest to the reader community.

Given my comments below, I would recommend publication of this manuscript with changes below, mainly due to high utility for the community and because the in silico methods and overall focus on intragenomic phasing of polymorphic variants for methods development is generally useful. Items 1-7 are addressable with text and items 8-9 are addressable with data but could also be addressable with text if more generally written in via referencing, discussion, and/or data.

Specifically, I would suggest the following changes for this manuscript:

1. Referencing of other full-length SMRT sequencing papers including examples like Wagner et al (27842515, 2016) and Prootakham et al (28573244, 2018) which also have similar intent using full length 16S reads and SMRT-Seq to essentially achieve the same goals of this manuscript other than the work on intragenomic resolved taxonomic information.
2. The introduction would also benefit by making mention of benefits of using full length 16S versus what is not mentioned in the manuscript, but is a superior method except for cost consideration – which is WGS using SMRT sequencing or other long read platforms. Ultimately even more information is gained by long read WGS, however full length 16S is clearly more cost effective. Examples of the LR-WGS metagenomics papers for discussion are: (Bankevich, Cell Systems 2018) – (Hiraoka, Nat Comm, 2019) – (Beaulorier, Nat Biotech 2018). When comparing, the authors focused

mainly on comparing to short read WGS, which is outdated and should be compared to long read WGS efforts.

3. For the in silico analysis, Supplementary Figure 2 is quite useful if possible to integrate into the main manuscript alongside main Figure 2. It just better represents the statements made throughout in a nice visual way.
4. The manuscript would benefit by pointing out more details on the number of CCS full length reads (coverage assessments) required to split specific intra- and inter-genomic copies which would benefit future research as it's stated loosely but fairly unclear in the main manuscript. Specifically mentioning this in the results section when discussing the 36-species mock community would be good information. Because the 16S approach relies on amplicons, it's important to state this in the context of CCS reads instead of just passes and accuracy alone. Supplementary Figure 9 addresses this but context on how it might change from system to system may be important.
5. In Supplementary Figure 6 (and discussion of results) when discussing homopolymer errors, making mention of if more read depth could overcome the homopolymer errors would be useful.
6. I agree that reference biases are an issue to deal with when using high resolution methods like the author proposes with full length 16S, which compounds database and reference errors and creates the need for full de novo approaches. More comments comparing to LR-WGS would be useful in the results and discussion around this point.
7. The Results comment from lines 210-213 about Figure 2b is unclear to me. How is the variable region dependence (as stated by using V1-V3 faulting variant calls in V6-V9) directly shown in that figure?
8. The bacteroides results are strong but, for instance, in Figure 3b, it would be useful to show LR-WGS alongside mWGS and the full length V1V9 data. Since this data was a low genome number metasample, this would be cost effective to generate and curious to readers given the momentum of LR-WGS today with cheaper means of data generation on Sequel 1, Sequel II and ONT.
9. It's unclear if any repeats (biologic or technical) were produced in the mock community study or the 4-sample bacteroides study. A duplicate technical run, at a minimum, would have been beneficial for knowledge of technical error (sequencing and amplification methods) and to address methods robustness.

Reviewers' comments:

Reviewer #1 (Remarks to the Author):

This manuscript by Johnson et al is a well written and well presented; it represents a substantial amount of computational work along with somewhat less in the way of original laboratory-based studies to compare (in silico) long and short-read 16S microbiome data for determining species- (and strain-) level composition. The authors correctly concluded that by sequencing the entire 16S gene using PacBio's circular consensus sequencing protocol (as opposed to Illumina-based short-read partial 16S gene sequencing) that it is possible in a far higher percentage of cases to obtain species- and in some cases strain-specific data. They authors also correctly point out that it is important to be aware of the fact that individual bacteria have multiple copies of the 16S operon, and that therefore counting each high quality sequence with a different SNP profile as a different strain (species) could inflate the complexity of the reported microbiome data.

Whereas these data and their analyses are state-of-the-art, and the results and interpretation are highly believable - they are essentially derivative of several recent papers in the microbiome space - particularly Earl et al Microbiome 2018 (which they reference). In Earl et al, not only do they reach essentially the same conclusions, but they also go much further and develop a full length 16S database (which is available to the public) for species-specific ID. Moreover, Earl et al analyze multiple (as opposed to a single) mock microbiomes - including one that is approximately 10 times the complexity (in terms of species numbers) that the current paper does - and they perform detailed analyses on a large set of novel specimens from the human nasopharynx (something not undertaken in the current manuscript).

Importantly, in the Earl et al paper they also compare, in silico, the resolving power of short and long read sequences with regard to species-specificity and reach essentially the same conclusions as that of the authors of the current paper. Thus, although this reviewer has no quarrel with the results presented, there is little that is truly novel in this paper, excepting perhaps their dwelling on the multiple 16S operon phenomenon (which was addressed in Earl et al in the construction of the full-length 16S database).

REPLY:

We thank the reviewer for their comments and for drawing attention to the recent study by Earl et al. (2018), who presented a comprehensive analysis of the utility of PacBio for full-length 16S gene sequencing.

There are key differences between our work and that of Earl et al. 2018. Most importantly, our manuscript addresses the utility of the 16S rRNA gene for species- and strain-level bacterial taxonomic resolution, with a particular focus on leveraging intra-genomic 16S gene copy variants for this purpose: Figure 2 presents a theoretical demonstration of the feasibility of this approach (via a comparison of E. coli strains); Figure 3 demonstrates that intra-genomic 16S gene copy variants can be resolved in vivo; Figure 4 now demonstrates that intra-genomic 16S gene copy variants are common within the human gut metagenome and have the potential to help discriminate between strains for many different species.

While Earl et al. present an interesting and detailed study, they do not address these points of interest. Specifically, Earl et al. make no mention of strain-level analysis and they make no mention of using intra-genomic copy variants as a method for improving taxonomic resolution. Their experimental investigation of intra-genomic copy variants is limited to Supplementary Figure 13B, and their discussion of this phenomenon is limited

to mentioning that it may contribute to the artificial inflation of taxon estimates based on Amplicon Sequence Variants (point (4) in the Discussion).

We agree with the reviewer that the idea behind our in silico analysis is not novel. However, we include this analysis in our manuscript in order to make the case that (contrary to what is frequently claimed) the sequencing sub-regions of the 16S gene is not adequate for species-level taxonomic quantification. In making this point, we also present the most comprehensive in silico comparison of 16S sub-regions to date. This is likely to provide a valuable resource for other researchers wishing to understand the potential resolution limitations and taxonomic biases associated with targeting particular variable regions (see in particular Supplementary Figure 2 and Supplementary Table 3).

We're grateful for the reviewer's comments and in response we have re-written parts of our manuscript and generated substantial new data to emphasize the novel aspects of our work. We hope they will agree that the revised manuscript presents novel and important results.

Reviewer #2 (Remarks to the Author):

Johnson et al. present in silico and sequencing data to support the prospect of full 16S gene sequencing using single molecule real-time sequencing methods of the 1500 bp region. Using these methods, the authors demonstrate strain level resolution of bacterial communities, both experimentally mixed and primary. Also, the authors spend considerable effort to show CCS accuracy of long reads enables accuracy to properly phase SNP level discordances between intra-genomic copies of 16S in a metagenomic system. This approach further underscores the need for intragenomic analyses approaches not achievable with current short read, targeted 16S approaches that only leverage specific variable regions for taxonomic analyses of complex samples today.

The molecular methods (primer design and sequencing) are all now fairly common methods from publications in the 2016-2018 era enabling full-length 16S sequencing using SMRT-Seq and have been applied broadly to various metagenomics communities since – to show similar findings. Where this paper is novel is in the specific focus on the intragenomic profiling of variation between 16S copies, which is often overlooked by other publications. The first major claim for the paper is more follow up to recapitulate the utility of full-length 16S sequencing relative to papers like Wagner et al, 2016 (BMC uBiology), whereas the second claims on intragenomic and strain level demonstrations are much more novel and of interest to the reader community.

Given my comments below, I would recommend publication of this manuscript with changes below, mainly due to high utility for the community and because the in silico methods and overall focus on intragenomic phasing of polymorphic variants for methods development is generally useful. Items 1-7 are addressable with text and items 8-9 are addressable with data but could also be addressable with text if more generally written in via referencing, discussion, and/or data.

Specifically, I would suggest the following changes for this manuscript:

1. Referencing of other full-length SMRT sequencing papers including examples like Wagner et al (27842515, 2016) and Prootakham et al (28573244, 2018) which also have similar intent using full length 16S reads and SMRT-Seq to essentially achieve the same goals of this manuscript other than the work on intragenomic resolved taxonomic information.

REPLY:

We thank the reviewer for these suggestions. We have expanded our discussion to include these and other references (lines 314-319) and to make clear that previous studies have evaluated PacBio CCS as a method for high-throughput sequencing of the full 16S gene.

We have also included additional text to make clear that our study is not an attempt to emulate these previous findings (lines 321-330). In doing so we have highlighted the novel aspects of our work that the reviewer alludes to.

2. The introduction would also benefit by making mention of benefits of using full length 16S versus what is not mentioned in the manuscript, but is a superior method except for cost consideration – which is WGS using SMRT sequencing or other long read platforms. Ultimately even more information is gained by long read WGS, however full length 16S is clearly more cost effective. Examples of the LR-WGS metagenomics papers for discussion are: (Bankevich, Cell Systems 2018) – (Hiraoka, Nat Comm, 2019) – (Beaulorier, Nat Biotech 2018).

When comparing, the authors focused mainly on comparing to short read WGS, which is outdated and should be compared to long read WGS efforts.

REPLY:

We again thank the reviewer for the suggestion. However, we note that short read WGS was included in our study only for benchmarking purposes, because it is widely acknowledged as the current gold standard for detecting bacteria in the gut at species level. (We have now clearly stated this in the manuscript in lines 224-225.)

We did not include long-read WGS sequencing in this study because our primary focus is the 16S gene, rather than evaluation of a particular sequencing approach (i.e. PacBio), or an evaluation of all the available sequence-based approaches for identifying microbial taxa (i.e. short vs long-read approaches).

We agree that LR-WGS is an exciting field. However, in response to this and other reviewers' comments we invested considerable effort into expanding our analysis of intra-genomic 16S gene polymorphisms (see additional data, text and figures). We therefore chose not to address LR-WGS in this study. We do not mention the citations listed above in order to avoid unnecessary length and to avoid detracting from the main focus of our manuscript.

3. For the in silico analysis, Supplementary Figure 2 is quite useful if possible to integrate into the main manuscript alongside main Figure 2. It just better represents the statements made throughout in a nice visual way.

REPLY:

We agree that the figure mentioned presents useful additional information. We have addressed this comment by including a simplified version of Supplementary Figure 2 as an additional panel in Figure 1.

4. The manuscript would benefit by pointing out more details on the number of CCS full length reads (coverage assessments) required to split specific intra- and inter-genomic copies which would benefit future research as it's stated loosely but fairly unclear in the main manuscript. Specifically mentioning this in the results section when discussing the 36-species mock community would be good information. Because the 16S approach relies on amplicons, it's important to state this in the context of CCS reads instead of just passes and accuracy alone. Supplementary Figure 9 addresses this but context on how it might change from system to system may be important.

REPLY:

We agree that understanding the exact number of CCS reads required to robustly detect intragenomic copy variants is of great interest. We have addressed this point in our expanded analysis of individually cultured bacteria. Specifically, we took advantage of replicate sequencing of the same sequence libraries to quantify the measurement error present in our estimates of SNP frequencies. We then looked for a relationship between measurement error and sequencing depth as the reviewer suggests. We found no relationship, indicating that, as few as 200 reads are sufficient to detect SNPs indicative of intragenomic copy variants. This suggestion is now addressed in lines 586-602 of the revised manuscript and in the data presented in Supplementary Table 6.

A more detailed examination of the number of reads required to reflect the 'true' SNP profiles (such as those shown in Fig. 2 for E. coli K-12 MG1655 16S and O157 Sakai) is complicated by the fact that we do not know the

true 16S gene sequence(s) expected for most of the bacteria included in this study. This is because many of the reference genomes available reference genomes for these species (which are required to predict expected SNP frequencies) were sequenced using short-read technologies and may hence present collapsed representations of each rRNA operon. Interestingly, this may be the reason for the lack of 16S gene diversity shown for Bacteroides strain mpk in Fig. 3d.

5. In Supplementary Figure 6 (and discussion of results) when discussing homopolymer errors, making mention of if more read depth could overcome the homopolymer errors would be useful.

REPLY:

Thank you for the suggestion, we have now addressed this point in lines 321-330 of the revised discussion.

6. I agree that reference biases are an issue to deal with when using high resolution methods like the author proposes with full length 16S, which compounds database and reference errors and creates the need for full de novo approaches. More comments comparing to LR-WGS would be useful in the results and discussion around this point.

REPLY:

We agree that long-read shotgun sequencing has the potential to improve taxonomic resolution. However, please see our responses to comment 2 (and 8 below). Given our decision to expand our analysis to foreground the novel aspects of this study (in particular, examining the potential for intra-genomic copy variants to distinguish species and strains), we have chosen not to address LR-WGS in this manuscript. We hope that the reviewer agrees with this decision.

As justification for our decision, we would also point out that for WGS to help with identifying species that suffer from inaccurate, or poor representation in reference database, those species must receive adequate coverage in a WGS dataset. As many poorly annotated taxa are typically sparse and present at low relative abundance, obtaining adequate coverage may require large sequencing depths and may therefore not be particularly cost effective. We would therefore argue that LR-WGS may be useful in this context, but may not be the ultimate solution.

7. The Results comment from lines 210-213 about Figure 2b is unclear to me. How is the variable region dependence (as stated by using V1-V3 faulting variant calls in V6-V9) directly shown in that figure?

REPLY:

Apologies for the confusion. In response to this comment we have revised Figure 2 to clarify the results presented. We have also modified the associated text (lines 208-211).

Our intention was to state that the SNPs that distinguish 16S rRNA gene copies found in strain O157 Sakai from those found in K-12 MG1655 can be found in multiple variable regions (V1, V2, V6, V7, V9). Therefore, to in order to maximise the ability to distinguish sequences originating from these two strains, it's necessary to sequence the entire 16S gene.

8. The bacteroides results are strong but, for instance, in Figure 3b, it would be useful to show LR-WGS alongside mWGS and the full length V1V9 data. Since this data was a low genome number metasample, this would be cost effective to generate and curious to readers given the momentum of LR-WGS today with cheaper means of data generation on Sequel 1, Sequel II and ONT.

REPLY:

As mentioned in response to previous comments, the primary focus of our revised manuscript is an evaluation of the 16S gene for species and strain-level taxonomic resolution in microbiome studies. The goal of including short-read shotgun sequencing was to provide a gold standard against which to evaluate the ability of full-length 16S to accurately identify Bacteroides species. While including LR-WGS would be very interesting, it would not contribute to this goal.

Inclusion of LR-WGS data also risks recasting this study as an evaluation of PacBio sequencing technology, which, as Reviewer #1 points out, is not novel for 16S gene sequencing.

We therefore respectfully argue that, while interesting, the proposed changes would not be the most cost/time efficient way to expand the relevance of our work. We have instead further investigated the potential for intragenomic variation in the 16S gene to characterize and discriminate between species and strains. Our decision was based on comments from both reviewers that the most novel aspect of our work lies in the intragenomic profiling of variation between 16S copies.

We hope the reviewer agrees that these additional data make the current study valuable and worthy of publication in Nature Communications.

9. It's unclear if any repeats (biologic or technical) were produced in the mock community study or the 4-sample bacteroides study. A duplicate technical run, at a minimum, would have been beneficial for knowledge of technical error (sequencing and amplification methods) and to address methods robustness.

REPLY:

Based on this suggestion, we have now included four repeats of the mock community sequencing (in our initial manuscript, these replicates were presented as a single pooled dataset). We now use these replicates to demonstrate that profiles representing intra-genomic 16S gene polymorphisms are reproducible (Supplementary Figure 9). We also show the reproducibility of the relationship between sequencing error and CCS pass number (Supplementary Figure 5a).

We have also addressed technical errors in our expanded analysis, which included the sequencing of individual isolates. Our estimates of the reproducibility of SNP frequency estimates are included in the revised text (lines 586-606), and summary figures are now included in the supplementary materials (Supplementary Figure 13).

Reviewers' comments:

Reviewer #2 (Remarks to the Author):

The reviewers have appropriately addressed all concerns relevant to my previous review criteria.

Specifically, the inclusion of the text to include conversation that this work is not meant to be a technologic comparison between 16S and long read WGS is appropriate in that the focus is now more centered on 16S rRNA and CNV based stratification of strain and species level identification using full length reads. The addition of other studies using the 16S data also strengthens this focused perspective, as revised.

The additional supplement in Figure 1 revisions alongside added text to discuss supplemental Figure 6 and revisions to Figure 2 make the presentation of the full length 16S data and any unique error profiles that would impact the alignments and/or proper taxonomic calls much clearer. I think these edits and additions address my previous concerns in that regard.

Last, the addition of edits addressing the splitting of the previously pooled data in the bactericides mock community study - into individual replicate data to better clarify Supplemental Figure 5a, 9, and 13 toward a better presentation of reproducibility and robustness of the technique. This makes the presentation of that allocation of data much clearer toward the data presented.

These edits address my original concerns and while I continue to think that targeted 16S and whole genome comparison is of interest to the community, I agree with the authors that the direct comparison shouldn't be a requirement to publish this work as to keep the manuscript centered, so these edits meet the criteria to move forward for consideration with the editor.

I have no additional concerns after reviewing this revised manuscript.

Reviewers' comments:

Reviewer #1 (Remarks to the Author):

This manuscript by Johnson et al is a well written and well presented; it represents a substantial amount of computational work along with somewhat less in the way of original laboratory-based studies to compare (in silico) long and short-read 16S microbiome data for determining species- (and strain-) level composition. The authors correctly concluded that by sequencing the entire 16S gene using PacBio's circular consensus sequencing protocol (as opposed to Illumina-based short-read partial 16S gene sequencing) that it is possible in a far higher percentage of cases to obtain species- and in some cases strain-specific data. They authors also correctly point out that it is important to be aware of the fact that individual bacteria have multiple copies of the 16S operon, and that therefore counting each high quality sequence with a different SNP profile as a different strain (species) could inflate the complexity of the reported microbiome data.

Whereas these data and their analyses are state-of-the-art, and the results and interpretation are highly believable - they are essentially derivative of several recent papers in the microbiome space - particularly Earl et al Microbiome 2018 (which they reference). In Earl et al, not only do they reach essentially the same conclusions, but they also go much further and develop a full length 16S database (which is available to the public) for species-specific ID. Moreover, Earl et al analyze multiple (as opposed to a single) mock microbiomes - including one that is approximately 10 times the complexity (in terms of species numbers) that the current paper does - and they perform detailed analyses on a large set of novel specimens from the human nasopharynx (something not undertaken in the current manuscript).

Importantly, in the Earl et al paper they also compare, in silico, the resolving power of short and long read sequences with regard to species-specificity and reach essentially the same conclusions as that of the authors of the current paper. Thus, although this reviewer has no quarrel with the results presented, there is little that is truly novel in this paper, excepting perhaps their dwelling on the multiple 16S operon phenomenon (which was addressed in Earl et al in the construction of the full-length 16S database.

REPLY:

We thank the reviewer for their comments and for drawing attention to the recent study by Earl et al. (2018), who presented a comprehensive analysis of the utility of PacBio for full-length 16S gene sequencing.

*There are key differences between our work and that of Earl et al. 2018. Most importantly, our manuscript addresses the utility of the 16S rRNA gene for species- and strain-level bacterial taxonomic resolution, with a particular focus on leveraging intra-genomic 16S gene copy variants for this purpose: Figure 2 presents a theoretical demonstration of the feasibility of this approach (via a comparison of *E. coli* strains); Figure 3 demonstrates that intra-genomic 16S gene copy variants can be resolved in vivo; Figure 4 now demonstrates that intra-genomic 16S gene copy variants are common within the human gut metagenome and have the potential to help discriminate between strains for many different species.*

While Earl et al. present an interesting and detailed study, they do not address these points of interest. Specifically, Earl et al. make no mention of strain-level analysis and they make no mention of using intra-genomic copy variants as a method for improving taxonomic resolution. Their experimental investigation of intra-genomic copy variants is limited to Supplementary Figure 13B, and their discussion of this phenomenon is limited

to mentioning that it may contribute to the artificial inflation of taxon estimates based on Amplicon Sequence Variants (point (4) in the Discussion).

We agree with the reviewer that the idea behind our in silico analysis is not novel. However, we include this analysis in our manuscript in order to make the case that (contrary to what is frequently claimed) the sequencing sub-regions of the 16S gene is not adequate for species-level taxonomic quantification. In making this point, we also present the most comprehensive in silico comparison of 16S sub-regions to date. This is likely to provide a valuable resource for other researchers wishing to understand the potential resolution limitations and taxonomic biases associated with targeting particular variable regions (see in particular Supplementary Figure 2 and Supplementary Table 3).

We're grateful for the reviewer's comments and in response we have re-written parts of our manuscript and generated substantial new data to emphasize the novel aspects of our work. We hope they will agree that the revised manuscript presents novel and important results.

Reviewer #2 (Remarks to the Author):

Johnson et al. present in silico and sequencing data to support the prospect of full 16S gene sequencing using single molecule real-time sequencing methods of the 1500 bp region. Using these methods, the authors demonstrate strain level resolution of bacterial communities, both experimentally mixed and primary. Also, the authors spend considerable effort to show CCS accuracy of long reads enables accuracy to properly phase SNP level discordances between intra-genomic copies of 16S in a metagenomic system. This approach further underscores the need for intragenomic analyses approaches not achievable with current short read, targeted 16S approaches that only leverage specific variable regions for taxonomic analyses of complex samples today.

The molecular methods (primer design and sequencing) are all now fairly common methods from publications in the 2016-2018 era enabling full-length 16S sequencing using SMRT-Seq and have been applied broadly to various metagenomics communities since – to show similar findings. Where this paper is novel is in the specific focus on the intragenomic profiling of variation between 16S copies, which is often overlooked by other publications. The first major claim for the paper is more follow up to recapitulate the utility of full-length 16S sequencing relative to papers like Wagner et al, 2016 (BMC uBiology), whereas the second claims on intragenomic and strain level demonstrations are much more novel and of interest to the reader community.

Given my comments below, I would recommend publication of this manuscript with changes below, mainly due to high utility for the community and because the in silico methods and overall focus on intragenomic phasing of polymorphic variants for methods development is generally useful. Items 1-7 are addressable with text and items 8-9 are addressable with data but could also be addressable with text if more generally written in via referencing, discussion, and/or data.

Specifically, I would suggest the following changes for this manuscript:

1. Referencing of other full-length SMRT sequencing papers including examples like Wagner et al (27842515, 2016) and Prootakham et al (28573244, 208) which also have similar intent using full length 16S reads and SMRT-Seq to essentially achieve the same goals of this manuscript other than the work on intragenomic resolved taxonomic information.

REPLY:

We thank the reviewer for these suggestions. We have expanded our discussion to include these and other references (lines 314-319) and to make clear that previous studies have evaluated PacBio CCS as a method for high-throughput sequencing of the full 16S gene.

We have also included additional text to make clear that our study is not an attempt to emulate these previous findings (lines 321-330). In doing so we have highlighted the novel aspects of our work that the reviewer alludes to.

2. The introduction would also benefit by making mention of benefits of using full length 16S versus what is not mentioned in the manuscript, but is a superior method except for cost consideration – which is WGS using SMRT sequencing or other long read platforms. Ultimately even more information is gained by long read WGS, however full length 16S is clearly more cost effective. Examples of the LR-WGS metagenomics papers for discussion are: (Bankevich, Cell Systems 2018) – (Hiraoka, Nat Comm, 2019) – (Beaulorier, Nat Biotech 2018).

When comparing, the authors focused mainly on comparing to short read WGS, which is outdated and should be compared to long read WGS efforts.

REPLY:

We again thank the reviewer for the suggestion. However, we note that short read WGS was included in our study only for benchmarking purposes, because it is widely acknowledged as the current gold standard for detecting bacteria in the gut at species level. (We have now clearly stated this in the manuscript in lines 224-225.)

We did not include long-read WGS sequencing in this study because our primary focus is the 16S gene, rather than evaluation of a particular sequencing approach (i.e. PacBio), or an evaluation of all the available sequence-based approaches for identifying microbial taxa (i.e. short vs long-read approaches).

We agree that LR-WGS is an exciting field. However, in response to this and other reviewers' comments we invested considerable effort into expanding our analysis of intra-genomic 16S gene polymorphisms (see additional data, text and figures). We therefore chose not to address LR-WGS in this study. We do not mention the citations listed above in order to avoid unnecessary length and to avoid detracting from the main focus of our manuscript.

3. For the in silico analysis, Supplementary Figure 2 is quite useful if possible to integrate into the main manuscript alongside main Figure 2. It just better represents the statements made throughout in a nice visual way.

REPLY:

We agree that the figure mentioned presents useful additional information. We have addressed this comment by including a simplified version of Supplementary Figure 2 as an additional panel in Figure 1.

4. The manuscript would benefit by pointing out more details on the number of CCS full length reads (coverage assessments) required to split specific intra- and inter-genomic copies which would benefit future research as it's stated loosely but fairly unclear in the main manuscript. Specifically mentioning this in the results section when discussing the 36-species mock community would be good information. Because the 16S approach relies on amplicons, it's important to state this in the context of CCS reads instead of just passes and accuracy alone. Supplementary Figure 9 addresses this but context on how it might change from system to system may be important.

REPLY:

We agree that understanding the exact number of CCS reads required to robustly detect intragenomic copy variants is of great interest. We have addressed this point in our expanded analysis of individually cultured bacteria. Specifically, we took advantage of replicate sequencing of the same sequence libraries to quantify the measurement error present in our estimates of SNP frequencies. We then looked for a relationship between measurement error and sequencing depth as the reviewer suggests. We found no relationship, indicating that, as few as 200 reads are sufficient to detect SNPs indicative of intragenomic copy variants. This suggestion is now addressed in lines 586-602 of the revised manuscript and in the data presented in Supplementary Table 6.

A more detailed examination of the number of reads required to reflect the 'true' SNP profiles (such as those shown in Fig. 2 for E. coli K-12 MG1655 16S and O157 Sakai) is complicated by the fact that we do not know the

true 16S gene sequence(s) expected for most of the bacteria included in this study. This is because many of the reference genomes available reference genomes for these species (which are required to predict expected SNP frequencies) were sequenced using short-read technologies and may hence present collapsed representations of each rRNA operon. Interestingly, this may be the reason for the lack of 16S gene diversity shown for *Bacteroides* strain mpk in Fig. 3d.

5. In Supplementary Figure 6 (and discussion of results) when discussing homopolymer errors, making mention of if more read depth could overcome the homopolymer errors would be useful.

REPLY:

Thank you for the suggestion, we have now addressed this point in lines 321-330 of the revised discussion.

6. I agree that reference biases are an issue to deal with when using high resolution methods like the author proposes with full length 16S, which compounds database and reference errors and creates the need for full de novo approaches. More comments comparing to LR-WGS would be useful in the results and discussion around this point.

REPLY:

We agree that long-read shotgun sequencing has the potential to improve taxonomic resolution. However, please see our responses to comment 2 (and 8 below). Given our decision to expand our analysis to foreground the novel aspects of this study (in particular, examining the potential for intra-genomic copy variants to distinguish species and strains), we have chosen not to address LR-WGS in this manuscript. We hope that the reviewer agrees with this decision.

As justification for our decision, we would also point out that for WGS to help with identifying species that suffer from inaccurate, or poor representation in reference database, those species must receive adequate coverage in a WGS dataset. As many poorly annotated taxa are typically sparse and present at low relative abundance, obtaining adequate coverage may require large sequencing depths and may therefore not be particularly cost effective. We would therefore argue that LR-WGS may be useful in this context, but may not be the ultimate solution.

7. The Results comment from lines 210-213 about Figure 2b is unclear to me. How is the variable region dependence (as stated by using V1-V3 faulting variant calls in V6-V9) directly shown in that figure?

REPLY:

Apologies for the confusion. In response to this comment we have revised Figure 2 to clarify the results presented. We have also modified the associated text (lines 208-211).

Our intention was to state that the SNPs that distinguish 16S rRNA gene copies found in strain O157 Sakai from those found in K-12 MG1655 can be found in multiple variable regions (V1, V2, V6, V7, V9). Therefore, to in order to maximise the ability to distinguish sequences originating from these two strains, it's necessary to sequence the entire 16S gene.

8. The bacteroides results are strong but, for instance, in Figure 3b, it would be useful to show LR-WGS alongside mWGS and the full length V1V9 data. Since this data was a low genome number metasample, this would be cost effective to generate and curious to readers given the momentum of LR-WGS today with cheaper means of data generation on Sequel 1, Sequel II and ONT.

REPLY:

As mentioned in response to previous comments, the primary focus of our revised manuscript is an evaluation of the 16S gene for species and strain-level taxonomic resolution in microbiome studies. The goal of including short-read shotgun sequencing was to provide a gold standard against which to evaluate the ability of full-length 16S to accurately identify Bacteroides species. While including LR-WGS would be very interesting, it would not contribute to this goal.

Inclusion of LR-WGS data also risks recasting this study as an evaluation of PacBio sequencing technology, which, as Reviewer #1 points out, is not novel for 16S gene sequencing.

We therefore respectfully argue that, while interesting, the proposed changes would not be the most cost/time efficient way to expand the relevance of our work. We have instead further investigated the potential for intragenomic variation in the 16S gene to characterize and discriminate between species and strains. Our decision was based on comments from both reviewers that the most novel aspect of our work lies in the intragenomic profiling of variation between 16S copies.

We hope the reviewer agrees that these additional data make the current study valuable and worthy of publication in Nature Communications.

9. It's unclear if any repeats (biologic or technical) were produced in the mock community study or the 4-sample bacteroides study. A duplicate technical run, at a minimum, would have been beneficial for knowledge of technical error (sequencing and amplification methods) and to address methods robustness.

REPLY:

Based on this suggestion, we have now included four repeats of the mock community sequencing (in our initial manuscript, these replicates were presented as a single pooled dataset). We now use these replicates to demonstrate that profiles representing intra-genomic 16S gene polymorphisms are reproducible (Supplementary Figure 9). We also show the reproducibility of the relationship between sequencing error and CCS pass number (Supplementary Figure 5a).

We have also addressed technical errors in our expanded analysis, which included the sequencing of individual isolates. Our estimates of the reproducibility of SNP frequency estimates are included in the revised text (lines 586-606), and summary figures are now included in the supplementary materials (Supplementary Figure 13).